# In Vitro Digestion and Intestinal Absorption of Mycotoxins Due to Exposure from Breakfast Cereals: Implications for Children’s Health

**DOI:** 10.3390/toxins16050205

**Published:** 2024-04-25

**Authors:** Soraia V. M. de Sá, Miguel A. Faria, José O. Fernandes, Sara C. Cunha

**Affiliations:** LAQV-REQUIMTE, Laboratory of Bromatology and Hydrology, Faculty of Pharmacy, University of Porto, Rua de Jorge Viterbo Ferreira, 228, 4050-313 Porto, Portugal; nano.soraia@gmail.com (S.V.M.d.S.); mfaria@ff.up.pt (M.A.F.); josefer@ff.up.pt (J.O.F.)

**Keywords:** breakfast cereals, mycotoxin co-occurrence, *in vitro* digestion, LC-MS/MS, children, bioaccessibility, intestinal absorption

## Abstract

Breakfast cereals play a crucial role in children’s diets, providing essential nutrients that are vital for their growth and development. Children are known to be more susceptible than adults to the harmful effects of food contaminants, with mycotoxins being a common concern in cereals. This study specifically investigated aflatoxin B1 (AFB1), enniatin B (ENNB), and sterigmatocystin (STG), three well-characterized mycotoxins found in cereals. The research aimed to address existing knowledge gaps by comprehensively evaluating the bioaccessibility and intestinal absorption of these three mycotoxins, both individually and in combination, when consumed with breakfast cereals and milk. The *in vitro* gastrointestinal method revealed patterns in the bioaccessibility of AFB1, ENNB, and STG. Overall, bioaccessibility increased as the food progressed from the stomach to the intestinal compartment, with the exception of ENNB, whose behavior differed depending on the type of milk. The ranking of overall bioaccessibility in different matrices was as follows: digested cereal > cereal with semi-skimmed milk > cereal with lactose-free milk > cereal with soy beverage. Bioaccessibility percentages varied considerably, ranging from 3.1% to 86.2% for AFB1, 1.5% to 59.3% for STG, and 0.6% to 98.2% for ENNB. Overall, the inclusion of milk in the ingested mixture had a greater impact on bioaccessibility compared to consuming the mycotoxins as a single compound or in combination. During intestinal transport, ENNB and STG exhibited the highest absorption rates when ingested together. This study highlights the importance of investigating the combined ingestion and transport of these mycotoxins to comprehensively assess their absorption and potential toxicity in humans, considering their frequent co-occurrence and the possibility of simultaneous exposure.

## 1. Introduction

Mycotoxin toxicity, bioaccessibility, and intestinal absorption form a critical triad for food safety, human health, and agricultural practices. Understanding these interlinked concepts is crucial to assess the impact of mycotoxins on human health and their path to the systemic circulation, where they exert toxic effects.

Produced by various molds, primarily *Aspergillus*, *Penicillium*, and *Fusarium* species, mycotoxins contaminate a wide range of agricultural products [1,2]. Notorious for their potent toxic effects on humans and animals, they can cause acute and chronic illnesses, immunosuppression, cancer, gastrointestinal disorders, and kidney damage [3]. Aflatoxins, a well-recognized group of mycotoxins known for their potent toxicity, include aflatoxin B1 (AFB1), the most prevalent and carcinogenic member classified as Group 1 by the International Agency for Research on Cancer (IARC) [4]. Aflatoxins are produced by *Aspergillus flavus* and *A. parasiticus*, typically found in soil, decaying vegetation, hay, and grains undergoing microbiological deterioration [5]. The European Food Safety Authority (EFSA) reports a persistent dietary intake of AFB1 in younger age groups, ranging from 0.08 to 1.78 ng/kg body weight per day on the lower end and from 0.58 to 6.95 ng/kg body weight per day on the upper end [3]. This persistent exposure underlines the importance of research on aflatoxin-related health effects in children [3].

Climate change contributes to the emergence of unregulated mycotoxins, such as enniatin B (ENNB) and sterigmatocystin (STG), which require consistent monitoring due to their potential harm [6]. These “emerging mycotoxins” have been detected in various food products like cereals, nuts, and processed foods, including breakfast cereals [7]. As a food highly consumed by children, breakfast cereals raise specific concerns. Children are more vulnerable to food contaminants due to the potential for substantial damage to intestinal enterocytes (cells lining the small intestine) by these toxins [8,9,10]. Therefore, studying the effects and fate of these compounds within the gastrointestinal tract is of paramount importance.

Concerning these compounds’ toxicity, STG serves as a precursor for aflatoxin production and shares structural similarities with it. It is produced by *Aspergillus* species, such as *A. versicolor* and *A. nidulans*, under optimal environmental conditions of 27–29 °C [11]. While STG is associated with low-level acute toxicity, the primary concern lies in its carcinogenic properties, roughly one-tenth of those of aflatoxin B1 [12]. ENNB belongs to a group of cyclic hexadepsipeptides produced by several *Fusarium* species such as *F. avenaceum*, *F. tricinctum*, *F. poae*, *F. sporotrichioides*, and *F. langsethiae* [13,14] and its contamination usually occurs pre-harvest [15]. ENNB exhibits strong cytotoxic effects, inducing cell death via mitochondrial damage [16]. Despite the EFSA asserting that short-term exposure to ENNB does not pose immediate health risks for humans, studies have demonstrated strong cytotoxic activity in different human cell lines, such as intestinal Caco-2 cells [17,18]. This suggests potential health risks with long-term exposure, warranting further investigation.

The quantity of ingested mycotoxin may not reflect the actual amount available for exerting its toxic effects. Only a fraction becomes bioaccessible within the gastrointestinal tract and further bioavailable upon reaching the bloodstream [19]. *In vitro* methodologies, such as simulated digestion models and intestinal absorption cellular models (e.g., Caco-2/HT29MTX co-culture), have gained traction in recent years [20,21,22]. These models mimic the human digestive and absorption process, allowing for researchers to investigate the influence of the food matrix on molecule bioavailability [19]. Cereal matrices have been extensively studied in this context, demonstrating high bioaccessibility for many toxins [23,24,25]. However, significant knowledge gaps remain. Although there are studies exploring ENNB bioaccessibility in breakfast cereals [24], and AFB1 bioaccessibility in plant-based milks [26], there is no existing literature on intestinal absorption of any of these compounds in Caco-2/HT29MTX co-cultures, with only one study focusing on the effects of AFB1’s metabolite (AFM1) within these co-cultures [27]. Although IARC monographs provide valuable information on AFB1’s *in vivo* absorption in humans [28,29], further research is necessary to understand the combined effects of these mycotoxins during intestinal absorption, particularly for ENNB and STG.

Given the limited research on mycotoxin bioaccessibility and intestinal absorption in children [28], particularly concerning combined ingestion, this study aimed to comprehensively assess these processes for AFB1, ENNB, and STG. This will enhance the accuracy of risk evaluation for mycotoxin exposure in this vulnerable population. 

This study employed a validated *in vitro* gastrointestinal model mimicking the human digestive system and a well-characterized intestinal absorption cellular model to comprehensively assess the bioavailability of target compounds. We investigated the bioaccessibility of AFB1, ENNB, and STG, both individually and in combination, in artificially contaminated breakfast cereals. We also examined their bioaccessibility in cereals with semi-skimmed milk (including milk alone) and their subsequent isolated and combined absorption in human-derived epithelial intestinal Caco-2/HT-29MTX co-culture cells. These co-culture cells represent the first human intestinal barrier encountered by mycotoxins after ingestion and digestion. Notably, this research appears to be the first to investigate the transport of these mycotoxins within this specific intestinal cell model (Caco-2/HT-29MTX co-culture).

## 2. Results and Discussion

### 2.1. Method Performance

Matrix calibration curves for digested matrices were generated to mitigate interferences stemming from matrix components. A linear response, with coefficients of correlation (r) exceeding 0.99, was observed for all mycotoxins in intestinal fraction of digested matrices. In the case of gastric fraction of digested matrices, favorable linear correlations (r > 0.98) were achieved only for ENNB (Appendix A). Validation of the method for AFB1 and STG in gastric fraction proved challenging due to persistent interferences that the clean-up process failed to eliminate. The limits of detection (LODs) ranged from 0.3–0.9 μg/L to 0.1–0.8 μg/L for gastric and intestinal fractions of digested matrices, respectively, and the limits of quantification (LOQs) ranged from 1.1–3.1 μg/L to 0.4–2.5 μg/L for gastric and intestinal fractions of digested matrices, respectively (Appendix A). Recovery percentages surpassed 70% for the four matrices, except for ENNB in breakfast cereal with semi-skimmed milk and in breakfast cereal with soy beverage, AFB1 in breakfast cereal with soy beverage, and STG in breakfast cereal with semi-skimmed lactose-free milk, which exhibited recoveries below 60%, and the coefficient of variation (%CV) values for interday and intraday precision were below 15% and 16%, respectively (Appendix A). 

Concerning transport assays, a linear response was verified for the three mycotoxins (with r > 0.98). The LODs and LOQs were observed to be in the ranges of 0.1 to 0.5 μg/L and 0.2 to 1.5 μg/L, respectively. The coefficient of variation (%CV) values for both interday and intraday precision were found to be below 16% (Appendix A).

### 2.2. Mycotoxins Bioaccessibility

Bioaccessibility values after *in vitro* digestion (gastric and intestinal) of AFB1, ENNB, and STG in breakfast cereals, and breakfast cereals with different milks and soy beverage are presented in Table 1. AFB1 and STG gastric bioaccessibilities were not evaluated, as already described in Section 2.1. Overall, an upward trend was noted throughout the gastrointestinal tract, indicating an increase in mycotoxin bioaccessibility from the gastric to the intestinal phase, except ENNB in the matrices with milk. The overall percentages of bioaccessibility varied within the following range accordingly to the matrix: very low to 98.2% for BC, very low to 47.2% for BCSSM, very low to 21.6% for BCSSMLF, and very low to 26.7% for BCSB. In terms of intestinal bioaccessibility, AFB1 demonstrated values ranging from 3.1% to 86.2%, STG exhibited values between 1.5% and 59.3%, and ENNB showed values spanning from 0.6% to 98.2%.

Concerning the matrix complexity, in general, when the breakfast cereals were consumed mixed with milk, all mycotoxins showed lower bioaccessibilities than when cereals were ingested alone. ENNB and STG presented the lowest percentages of the bioaccessible fractions (less than 2%), namely, when in mixture.

The bioaccessibility of mycotoxins has been reported to be influenced by various factors, including its chemical structure, pH during the digestion process, and the composition of food matrices [25]. The disparities in bioaccessibility (%) presented in Table 1 can also be elucidated by the composition of the digested food matrices as referred to in previous research for both nutrients and toxic compounds [30]. Cereal-based foods typically contain natural compounds, such as adsorbent dietary fibers, which interact with mycotoxins, resulting in a reduction in their bioaccessibility [31]. Interactions with other food components can influence mycotoxin bioaccessibility, with mycotoxins demonstrating the ability to bind primarily with proteins and lipids within the food matrix [24]. Moreover, dietary fibers have already been used as a protective measure against mycotoxicosis by their inclusion in food and feed products, offering a cost-effective method for detoxification [32,33]. Table 3 (Section 4) displays the proximate nutritional composition of each product used in the gastrointestinal digestion process. The bioaccessibility values for ENNB, in the breakfast cereal digested sample (corn with honey), are 91.2% and 98.2% for the mycotoxin in isolated form and in the mixture, respectively. These values surpass those reported in the only study in a similar type of digested matrix (corn flakes) by Prosperini et al. (ranging from 43 to 70%) [24]. These differences can be attributed to the composition of our sample. Although having the same fiber content (3 g/100 g of product), it exhibits lower levels of fat (0.6 g against 0.7 g/100 g of product) and protein (5.5 g against 8.0 g/100 g of product), which enables greater bioaccessibility, given that the levels of fat and proteins are lower, resulting in less retention of mycotoxins in the matrix, making them more bioaccessible.

For the digested breakfast cereal samples with the different types of milk and soy beverage, as can be seen in Table 1, the bioaccessibility percentages of mycotoxins are much lower compared to the digested breakfast cereal sample alone. This behavior could be explained by the fact that the milks and the soy beverage contain a higher fat content (>3 g/100 mL) (Table 3), which leads to lower bioaccessibility. The high fat content can undergo the release of mycotoxins from the matrix, most probably related to the different lipophilic and hydrophilic properties of these molecules [34]. ENNB demonstrates to be the most lipophilic (log P_o/w_ of 3.61), being poorly soluble in water, followed by STG and AFB1 (log P_o/w_ of 2.61 and 2.09, respectively), moderately hydrophilic, which explains the greater retention of these compounds in matrices containing milk. Additionally, the digested breakfast cereal with soy beverage exhibits even lower values compared to the other matrices with milk. One of the reasons for this decrease in bioaccessibility, with the exception of ENNB, is due to the soy beverage’s increased fiber content, which is not found in the other types of milk.

### 2.3. Cell Monolayer Integrity Control

To validate that the transport study adhered to recommended conditions, two distinct measurements were employed to assess monolayer integrity after exposure for 180 min to the detoxified bioaccessible fraction and a mixed standard solution of mycotoxins: (i) the determination of mycotoxin cytotoxicity with the concentrations under investigation and (ii) the measurement of trans-epithelial electrical resistance (TEER). TEER determination furnishes details regarding the consistency of the Caco-2 cell layer on the filter support and the integrity of the tight junctions established between the polarized cells [22], and a decline in that TEER values could indicate an increase in the permeability of the tight junctions due to cell detachment, etc. [35]. TEER values for Caco-2/HT-29MTX monolayers were complying with values reported in the literature [22]. No significant differences (*p* < 0.05) were observed in TEER values, measured before (0 h, ≈1390 Ωcm^2^) and after transport experiments (3 h, ≈1730 Ωcm^2^), following exposure to mycotoxins, either individually or in a mixture in both matrices. 

The detoxified bioaccessible fraction was serially diluted in the transport medium (HBSS) and tested by exposing it to the monolayers for 180 min, as well as a standard mixture of the contaminants in relevant concentrations ENNB (0.31 µM), STG (0.62 µM), and AFB1 (0.64 µM). It is worth noting that these concentrations are within the range found in food [2,36], despite the low concentrations of AFB1 reported in the literature and the stringent regulations governing its maximum content in food [37]. The bioaccessible fraction in a dilution higher than 6× did not impart cell toxicity to the co-cultured cells Caco-2/HT-29MTX; thus, this dilution was selected for further assays. The toxicity of the compounds themselves was also not noticed after 180 min of exposure, thereby not affecting cell viability (Figure 1) as measured by the MTT assay. These preliminary assays assured that the monolayer integrity of differentiated Caco-2/HT-29MTX cells remained uncompromised during transport assays.

### 2.4. Intestinal In Vitro Absorption Assays

The apical–basolateral transport of individual AFB1, ENNB, STG, and their combined mixture was evaluated across Caco-2/HT-29MTX monolayers. Human cell models, such as the Caco-2/HT-29MTX co-culture model, serve as prominent tools for examining intestinal transport and absorption [22]. This model, involving the co-culture of Caco-2 and HT-29MTX, proves valuable for exploring transport across the intestinal epithelium and studying bacterial adhesion and invasion [22]. The inclusion of the mucin layer by HT-29MTX and the assessment of cell layer permeability are pivotal for such investigations, and the co-culture approach yields results that align more closely with the *in vivo* conditions compared to monocultures [22].

Figure 2 presents the outcomes related to the transport across Caco-2/HT-29MTX monolayers of each mycotoxin individually and in a mixture, expressed as mass percentage transported over time, in both cereal and cereal with semi-skimmed milk matrices. All mycotoxins under study were absorbed; however, varied transport rates were noted based on the mycotoxin type, the presence of another molecule (isolated or mixed), and the type of matrix. In theory, the cumulative fraction of a passively transported drug across Caco-2 cell monolayers should exhibit a linear increase over time when conducted under sink conditions. However, in practical experiments, deviations from this scenario may occur, leading to non-linear transport behavior.

AFB1 exhibited an almost linear transfer to the receiver compartment (transport percentages similar to theoretical values until 60 min) when isolated in the breakfast cereal matrix, resulting in high transport rates of around 87% (Figure 2A). When in a mixture, the transport rate was much lower and in accordance with theoretical values, achieving maximum percentages of around 7%. In contrast, in the presence of semi-skimmed milk, AFB1 showed a linear and slower rate of transport, reaching 13% and 14%, when isolated and in mixture, respectively, being in accordance with theoretical values (Figure 2B). This implies that the addition of milk to the matrix and the combination with other mycotoxins may reduce the intestinal transport of AFB1.

The ENNB transport profile remained consistent across all situations, following the same trend regardless of the matrix or whether ENNB was isolated or in a mixture. ENNB isolated demonstrated a similar trend, whether in breakfast cereal alone (Figure 2C) or with semi-skimmed milk (Figure 2D); the same trend was verified when ENNB was in mixture, achieving maximum percentages of transport around 80 and 72% when isolated and lower ones, around 53 and 62%, when combined. This decrease in transport percentage suggests that combination with other mycotoxins may hinder the intestinal transport of ENNB, as verified already for AFB1.

In the case of STG, the transport rate showed linearity in the breakfast cereal with semi-skimmed milk, rapidly transferred to the receiver compartment whether transported in isolation or in a mixture, resulting in very high transport percentages (92 and 96%, respectively), and only displaying values similar to theoretical transport up to 60 min (Figure 2F). When absorption occurred only in the breakfast cereal matrix (Figure 2E), a slower rate was observed until 120 min, followed by a more rapid transport until 180 min, achieving maximum transport percentages of 38 and 28%, whether isolated or in a mixture, respectively. This suggests that the addition of milk to the matrix may increase the intestinal transport of STG, when the molecule is not in the presence of AFB1 and ENNB.

The absorption of mycotoxins in the intestine is a complex process influenced by several factors, including the type of molecule, the presence of other substances, and the physiological conditions. When mycotoxins are present in a mixture, several mechanisms may contribute to a decrease in their intestinal absorption. As mentioned before, the food matrix can affect their absorption, due to the presence of other food components, such as fats, carbohydrates, proteins, fibers, and various other nutrients, that may influence the solubility and absorption of mycotoxins. Moreover, different mycotoxins may compete for the same transporters or binding sites in the intestine. If these binding sites become saturated with one mycotoxin, it may reduce the absorption of others, leading to a lower overall absorption rate [38].

In addition to detailing the mycotoxin transport over time, we calculated the apparent permeability (P_app_) to provide a precise estimation of mycotoxin intestinal absorption. As shown in Figure 2, overall, all mycotoxins exhibited rapid initial absorption, followed by a reduction. This suggests that during the assay, sink conditions were not always adequately confirmed, deviating from a linear fit of mycotoxin content [39]. Consequently, permeability values were computed for non-sink conditions using the equation outlined in Section 4.6 [39,40]. According to Tavelin et al., 2002, P_app_ values obtained under this condition align more closely with the actual permeability coefficients of epithelial cell monolayers in vivo. In Table 2, we present the calculated P_app_ values derived from the curves depicting the transport of mycotoxins, as illustrated in Figure 2.

Artursson et al., 2001 suggested that permeability coefficients exceeding 1 × 10^−6^ cm/s indicate high permeability [41]. Therefore, the P_app_ values derived for mycotoxins imply that AFB1, ENNB, and STG were effectively absorbed in Caco-2/HT-29MTX cells, whether individually or in combination. AFB1 exhibits the highest apparent permeability value in the breakfast cereal matrix when isolated, followed by ENNB and finally STG. However, in the cereals with milk, AFB1 shows the lowest apparent permeability value, with ENNB having the highest. Regarding the permeability values when mycotoxins are in a mixture, in cereal matrix, ENNB demonstrates the highest value, while AFB1 has the lowest result. Overall, the simultaneous transport of mycotoxins led to a significant decrease in the P_app_, with the exception of STG in breakfast cereal with semi-skimmed milk matrix (Table 2).

Our results indicate differences in the P_app_ of mycotoxins between the two matrices. The presence of milk alongside breakfast cereals demonstrated a notable impact on the absorption kinetics. This finding suggests that the matrix composition plays a crucial role in modulating mycotoxin bioavailability, and the inclusion of milk can be a contributing factor. The observed variations in P_app_ values between the breakfast cereal only and breakfast cereal with milk matrices may be attributed to the complex interplay of components present in milk, as stated above for bioaccessibility differences. Milk contains various bioactive molecules, such as proteins, fats, and carbohydrates, which could potentially interact with mycotoxins, affecting their transport across the intestinal barrier. The specific mechanisms behind these interactions warrant further investigation to fully understand the dynamics of mycotoxin absorption in the presence of milk.

Concerning the mass balance values in both matrices (i.e., the total mycotoxin recovered from the apical and basolateral compartments at the final of the experiment divided by the initial amount in the apical compartment), they varied between 28 and 156% for breakfast cereals and 29 and 172% for breakfast cereals with semi-skimmed milk (Table 2). These values, namely, the lower ones, could be due to some reasons, such as the mycotoxin having been adsorbed to the experimental apparatus (such as the plastic of the well plate, the insert device, or the membrane itself), the compound being held within the cells or in the cell membranes, undergoing metabolic processes, compound precipitation, or degradation during incubation [40,42].

The Caco-2 cell model is frequently employed for predicting the human fraction absorbed (FA%), which is the fraction of a drug absorbed in humans following oral administration [43]. This is achieved by establishing correlations between the apparent permeabilities in the apical-to-basolateral (AB) direction across Caco-2 cell monolayers of molecules and the experimental human fraction absorbed data. The outcome is a sigmoidal relationship between the human fraction absorbed and the logarithm of the apparent permeability (log P_app_) of molecules [42]. Therefore, by establishing this correlation, the *in vitro* permeability of a compound in Caco-2 cells can serve as a predictive measure for human absorption. Besides being validated by *in vivo* comparison using several drugs, FA% values are deemed more intuitive than P_app_ values when comprehending the intestinal absorption of compounds, providing the added advantage of categorizing permeability into low (0–20%), medium (20–80%), and high (80–100%) absorption. Figure 3 illustrates the FA% of each of the assayed conditions in both matrices under study, showcasing their position on the sigmoidal curve based on their percentage of absorption. AFB1 exhibits the highest FA% when isolated, with a percentage of 98.8% in breakfast cereal but a lower value of 82.6% in breakfast cereal with milk. Following this, ENNB demonstrates FA% values of 98.6% and 98.7% (breakfast cereal and breakfast cereal with milk, respectively), while STG ranges between 96.7% (breakfast cereal) and 98.5% (breakfast cereal with milk). Concerning the transport of mycotoxins in a mixture, STG attains the highest FA% value of 98.9% in breakfast cereal with milk, slightly lower at 94.9% in breakfast cereal alone. ENNB follows with values of 98.5% (breakfast cereal) and 98.4% (breakfast cereal with milk), and AFB1 exhibits percentages of 66.1% and 84.2%, respectively. Consequently, all mycotoxins are categorized as highly absorbed, whether isolated or in a mixture, in both matrices under study. The only exception is AFB1 in the mixture within the breakfast cereal matrix, showing a medium level of absorption (Figure 3). This can lead us to the conclusion that overall, the effect of the milk matrix did not influence the final correspondent predicted FA% in humans, being, overall, all the analyzed molecules highly absorbed in the gastrointestinal tract. On the other side, the presence of other mycotoxins did have an impact in AFB1 absorption, leading to a relevant decrease in absorption from high to medium. Extrapolating P_app_ values to FA% revealed that only substantial differences in P_app_ values significantly impact a compound’s *in vivo* absorption.

Transport mechanisms across absorptive epithelia can manifest through various routes, including passive pathways such as transcellular and/or paracellular transport, active transcellular transport mediated by transporters, or transcytosis [38,39]. Mycotoxins are absorbed through the small intestine primarily via passive diffusion, exhibiting a remarkably high absorption rate [44]. The available data (*in vivo* rat whole small intestine) indicate that nearly total absorption of AFB1 can occur within the intestinal tract [45]. Nevertheless, there is a lack of existing literature on the intestinal absorption of AFB1, ENNB, and STG in Caco-2/HT29MTX cell co-cultures and in the presence of the digested food matrix. Only one study has been conducted, focusing on the impact of AFB1 metabolite (AFM1) in these co-cultures [27], and one study *in vitro* using Caco-2 cells for AFB1 [46]; however, numerous studies have been dedicated to utilizing Caco-2 cells for assessing the absorption and toxicity of other crucial mycotoxins like trichothecenes and zearalenone [47,48,49]. Therefore, this study is highly significant as it aims to contribute with more data, enabling a more precise determination of risk assessment.

When simultaneously considering values of both intestinal bioaccessibility and predicted fraction absorbed, in percentage, data show (Figure 4) that the inclusion of milk imparts an overall decrease in bioaccessibility (all below 50%) and AFB1 appears as the least absorbed (although still in high percentage, as stated above) with the exception of the case in which it is ingested in the cereal matrix and as a single compound. Overall, the effect of the inclusion of milk in the ingested matrix has more impact than substances ingested as a mixture or single mycotoxin. When taken together the bioaccessibility and FA%, which can be correlated with the final bioavailability, it can be verified that the most bioaccessible and absorbed toxin is ENNB (ingested with cereals only) and that the least bioavailable is AFB1 ingested cereals with milk.

## 3. Conclusions

This study emphasizes the importance of examining the combined ingestion and transport of mycotoxins to comprehensively understand their absorption and more accurately assess human exposure. This is particularly crucial considering the frequent co-occurrence and potential for simultaneous exposure to these toxins in breakfast cereals.

The *in vitro* gastrointestinal method revealed varying patterns of bioaccessibility for AFB1, ENNB, and STG in breakfast cereals, including those consumed with different milks and soy beverages. An overall upward trend in mycotoxin bioaccessibility was observed from the gastric to intestinal stages, except for ENNB, which showed variable behavior depending on the milk type. The overall bioaccessibility ranking, from highest to lowest, was digested breakfast cereal, breakfast cereal with semi-skimmed milk, breakfast cereal with lactose-free milk, and breakfast cereal with soy beverage. Bioaccessibility percentages varied widely across all samples and conditions, with AFB1 ranging from 3.1% to 86.2%, STG from 1.5% to 59.3%, and ENNB from 0.6% to 98.2%. Interestingly, ENNB displayed the highest bioaccessibility values, regardless of whether it was present alone or in combination.

Regarding intestinal transport, ENNB and STG exhibited the highest absorption rates when present together in a mixture, irrespective of the matrix. When isolated, their absorption rates were even higher, particularly in the breakfast cereal with milk matrix. Notably, AFB1 demonstrated the highest absorption rate within the isolated breakfast cereal matrix.

This study’s findings are significant because they provide novel insights into the bioaccessibility and intestinal absorption of mycotoxins in breakfast cereals. It uniquely considered both bioaccessibility and intestinal transport simultaneously within a real-world food matrix, mimicking realistic consumption scenarios. This information can be used to develop strategies for mitigating mycotoxin exposure risk in children. For instance, parents could be advised to choose cereals with milk, potentially reducing exposure, particularly to the highly toxic and prevalent AFB1.

## 4. Materials and Methods

### 4.1. Reagents and Materials

AFB1 (10 mg, >98% purity) was purchased from LGC (Teddington, Middlesex, UK). ENNB (1 mg, ≥95% purity), porcine α-amylase, pepsin, and bile and pancreatin extracts were all purchased from Sigma-Aldrich corp. (St. Louis, MO, USA). STG (1 mg, ≥98% purity) was purchased from BioViotica (Liestal, Switzerland). Acetonitrile (ACN), methanol (MeOH), and acetic and formic acids of high-performance liquid chromatography (HPLC) grade, as well as ammonium acetate (P.A.), were obtained from Merck (Darmstadt, Germany). Anhydrous magnesium sulphate (MgSO_4_) and bovine serum albumin (BSA) were purchased from Sigma-Aldrich and sodium chloride (NaCl) from VWR (Stříbrná Skalice, Czech Republic); both were calcinated at 500 °C for 5 h before use. Ultrapure water, purified with a “Seral” system (SeralPur Pro 90 CN), was used. MTT (3-(4,5-dimethylthiazol-2-yl)-2,5-diphenyltetrazolium bromide) and dimethyl sulfoxide (DMSO) were purchased from Duchefa Biochemie (Haarlem, The Netherlands). Fetal bovine serum (FBS), 0.25% trypsin solution, minimum essential medium non-essential amino acids (MEM NEAA) 100×, GlutaMAX™ 100×, and penicillin/streptomycin 100× solution (10,000 Units mL^−1^/10 mg mL^−1^) were all purchased from Gibco/Life technology corporation (Paisley, UK). High-glucose Dulbecco’s modified Eagle’s medium (DMEM) and Hanks Balanced Salt Solution (HBSS) were purchased from Biowest (Nuaillé, France). Syringe filters (PES, 0.22 μm and 0.45 μm pore) were purchased from TPP (Zollstarsse, Switzerland). Transwell inserts and plates (PS/PET membrane, 24 mm diameter, 0.4 μm pore size) for the transport assay were obtained from cellQART (Northeim, Germany). Standard stock solutions of each mycotoxin at 10 and 100 mg/L were prepared by diluting them in methanol for LC–MS/MS validation purposes and spiking of samples, respectively. Stock internal solutions of OTA-d5 (500 mg/L) and ^13^C_18_-STG (25 mg/L) were prepared in DMSO and ACN, respectively. All standard solutions were stored at −18 °C.

### 4.2. Cell Culture

Human-derived intestinal cells Caco-2 and HT-29MTX were kindly provided by the Fisico-Química Molecular group from University of Coimbra and from Faculty of Sciences University of Porto, respectively.

Cells were grown in 75 cm^2^ flasks, at 37 °C with 5% CO_2_, in complete medium (CM) with the following composition [22]: DMEM with 10% heat inactivate FBS, 1% penicillin/streptomycin, 1% non-essential amino acids (NEAA), and 1% glutamax. Cytotoxicity and permeability assays were performed under passages 73–76 and 74–77 for HT-29MTX and Caco-2 cells, respectively.

### 4.3. Spiking Samples

Before digestion, 16 μL of a 100 mg/L solution of each mycotoxin (400 µg/L), AFB1, ENNB, and STG, was used to spike 4 g of breakfast cereal homogeneously and left to incubate for 15 min at room temperature. Then, 16.67 mL of the different milks (semi-skimmed, semi-skimmed lactose free, and soy vegetable drink) was added. Each mixture was performed in triplicate (5 g each). The assay with only breakfast cereals (5 g) was spiked with 20 μL of a 100 mg/L solution of each mycotoxin (400 µg/L), homogeneously, and left to incubate during 15 min at room temperature.

The nutritional compositions of the previously mentioned breakfast cereal, milk, and soy beverage samples are presented in Table 3.

### 4.4. In Vitro Digestion

The *in vitro* digestion procedure was performed according to the internationally standardized method described by INFOGEST 2.0 [50]. Briefly, 5 g of sample (breakfast cereals and breakfast cereals with milk (semi-skimmed, semi-skimmed lactose-free, and soy vegetable drink)) was mixed with 4 mL of simulated salivary fluid (SSF), 0.5 mL of α-amylase solution at 6.04 U/mg in water, 25 μL 0.3 M CaCl_2_ solution, and 475 μL water. After 2 min incubation, the mixture was mixed with 8 mL of simulated gastric fluid (SGF), 5 μL 0.3 M CaCl_2_, 35 μL (in the case of breakfast cereals samples) or 88 μL (in the case of breakfast cereals with milks) of 6 M HCl to adjust to pH = 3, 1.460 mL (in the case of breakfast cereals samples) or 1.407 mL (in the case of breakfast cereals with milks) of water, and 0.5 mL of pepsin (2668.2 U/mg). The gastric mixture was then incubated at 37 °C for 120 min in an orbital shaker–incubator (ES-20, BioSan, Riga, Latvia) with integrated horizontal shaker at 250 rpm. Then, the gastric chime (10 mL) was mixed with 4.25 mL of simulated intestinal fluid (SIF) solution, 20 μL 0.3 M CaCl_2_, 40 μL (in the case of breakfast cereals samples) or 75 μL (in the case of breakfast cereals with milks) of 1 M NaOH to adjust the pH to 7.0, 1.940 mL (in the case of breakfast cereals samples) or 1.905 mL (in the case of breakfast cereals with milks) of water, 2.5 mL of pancreatin solution (4.7 U/mg based on trypsin activity), and 1.25 mL of bile solution (2.151 mmol/g). All digestions were made in triplicates plus one blank sample used to adjust the pH and used as a matrix for the LC–MS/MS calibration assays as well as for transport assays. After digestion, the samples were immediately centrifuged at 3000× *g* for 5 min to obtain the bioaccessible fraction (supernatant). Supernatants were then frozen at −20 °C until further analysis.

### 4.5. Cytotoxic Assay

The MTT assay was conducted to assess the impact on cell viability to the tested toxicants prepared in control bioaccessible fractions from *in vitro* digestion after suffering a heat treatment at 98 °C for 5 min then filtered throughout 0.45 and 0.22 µm filters and diluted with HBSS. Proliferating Caco-2 (45,000 cells/mL) and HT-29MTX (5000 cells/mL) were seeded in 96-well plates (TPP; Trasadingen, Switzerland), allowing one week for cell adherence at 37 °C, 5% CO_2_, and then exposed to mycotoxins at the concentrations used for transport assay: 0.31 µM (ENNB), 0.62 µM (STG), and 0.64 µM (AFB1) for 3 h. Cells treated with CM alone, and with HBSS, were used as a control on cell viability during the experiments. The medium was removed, after 180 min of exposure, and 100 μL of freshly prepared MTT solution (0.5 mg/mL) was added to each well and the plates placed in the incubator at 37 °C, 5% CO_2_ for 20 min, and after that was added to each well 100 μL of DMSO and let the plates rest for another 30 min before absorbance reading at 570 nm using an absorbance plate reader (SPECTROstar Nano, BMG Labtech, Offenburg, Germany). Results were expressed as % of cell viability.

### 4.6. Transport Assay

Caco-2/HT-29MTX co-culture [22] cells were seeded at a density of 1 × 10^5^ cells/cm^2^ (ratio 90:10, Caco-2/HT-29MTX) in 24 mm 6-well Transwell inserts with a pore size of 0.4 μm and a growth area of 4.5 cm^2^ (cellQART^®^, Northeim, Germany—ref.: 9300404). Tests were carried out with 2.5 mL in the basolateral compartment and 1.5 mL in the apical compartment. During the differentiation process, the culture medium was changed every two days, and the cells were used for the transport assay on the 26th day after seeding. For the transport experiments, the culture medium of the Caco-2/HT-29MTX co-culture cells was replaced with HBSS containing 0.5% (*w*/*v*) BSA in the basolateral compartment. BSA was added to minimizing the binding of compounds to the plastic surfaces [40].

Mycotoxins at the initial concentrations of 0.31 µM (ENNB), 0.62 µM (STG), and 0.64 µM (AFB1) (200 µg/L each) each were prepared in control bioaccessible fractions (digested samples without mycotoxins) from *in vitro* digestion after suffering a heat-shock treatment at 98 °C for 5 min, to inhibit the enzymes in order to maintain the cells’ viability [51], then filtered through 0.45 and 0.22 µm filters, sequentially, diluted with HBSS 1:8, and introduced in the apical compartments isolated or in mixture. At time points of 15, 30, 60, 120, and 180 min of absorption, 200 μL samples were taken from the basolateral compartments, and the same volume (200 μL) of HBSS was added to maintain the volume. The concentration of mycotoxins used for the absorption assay was selected to allow for the quantification of mycotoxins at all stages of the transport assay. The trans-epithelial electrical resistance (TEER) values were measured at 37 °C using a Millicell ERS-2 Voltohmmeter (Merck Millipore, Darmstadt, Germany) at the beginning and end of the experiment after washing the cell monolayer with HBSS to assess barrier integrity. The resistance was expressed in Ωcm^2^, calculated by multiplying the cell monolayer resistance (Ω) by the filter area (cm^2^). All transport assays were conducted in triplicate.

The assessment of mycotoxin intestinal permeability involved measuring transport rates across Caco-2/HT-29MTX cell monolayers, as outlined in previous studies [40]. The permeability coefficient (P_app_) in the apical → basolateral direction was determined using the following equation, specifically designed for experiments conducted under non-sink conditions [39,40]:$$C_{R}\left(t\right)=\frac{M}{V_{D}+V_{R}}+\left(C_{R\left(0\right)}-\frac{M}{V_{D}+V_{R}}\right)e^{-\mathrm{PappA}\left(\frac{1}{\mathrm{VD}}+\frac{1}{\mathrm{VR}}\right)t}$$

In this context, C_R_(t) represents the mycotoxin concentration in the receiver compartment over time, M denotes the total mycotoxin amount within the system, V_D_ and V_R_ stand for the volumes of the donor and receiver compartments, respectively. C_R0_ signifies the initial concentration of the mycotoxin in the receiver compartment at the onset of the time interval, A represents the surface area of the filter, and t signifies the elapsed time from the commencement of the interval. The permeability coefficient (P_app_) was derived through non-linear regression [39]. The mass balance was calculated through the following equation:$$\mathrm{Mass}\;\mathrm{balance}\;\left(\%\right)=\frac{\left[\left(V_{R}\times {\;C}_{R\left(\mathrm{final}\right)}\right)+\left(V_{D}\times {\;C}_{D\left(\mathrm{final}\right)}\right)\right]}{V_{D\left(0\right)}\times {\;C}_{D\left(0\right)}}\times 100$$

C_R_ and C_D_ represent concentrations on the receiver (R) and donor (D) sides of the monolayer at the experiment’s onset (0) and conclusion (final), with V denoting the respective volumes. The apical-to-basolateral permeability (P_app_) data obtained from Caco-2/HT-29MTX cells were employed to calculate the human fraction absorbed, FA (%), using the non-linear regression model outlined by Skolnik et al., 2010 [42].
$$\mathrm{FA}\;\left(\%\right)=\frac{100}{1+e^{\left(\left(-5.74-\mathrm{PappX}\right)/0.39\right)}}$$

In this context, 100 is defined as the sum of the minimum and maximum values of % FA, restricted within the range of 1 to 100%. The log P_appA→B_ value at 50% absorption in humans is represented as −5.74, while P_appX_ signifies the log P_appA→B_ for mycotoxins in Caco-2/HT-29MTX cells, as determined in the current study. Additionally, 0.39 corresponds to the slope derived from the model fit.

### 4.7. Mycotoxins Extraction and Cleanup

Bioaccessible and Bioavailable Fractions

Two hundred microliters (200 μL) of digested sample obtained as described in Section 2.3 were transferred into a conical microtube and a fixed concentration of OTA-d5 (40 μg/L) was added. Thereafter, 200 μL acidified ACN with 5% formic acid (*v*/*v*) were added along with 70 mg of MgSO_4_ anhydrous salt and 10 mg of NaCl and the tube was immediately vortexed for 10 s to prevent agglomeration of the salts. The tubes were then centrifuged at 13,000 rpm for 5 min to induce phase separation and mycotoxins partitioning. The organic phase was transferred to a 2 mL vial, evaporated to dryness under a stream of nitrogen, and finally reconstituted in 150 μL of mobile phase B (Section 4.5) and analyzed by LC–MS/MS. Each sample was injected three times.

### 4.8. Instrument and Analytical Conditions

MS/MS analysis was performed on a Quattro Micro triple quadrupole mass spectrometer (Waters, Manchester, UK) interfaced with a high-performance liquid chromatography (HPLC) system Waters Alliance 2695 (Waters, Milford). A Kinetex C18 2.6 µm particle size analytical column (150 × 4.6 mm) with pre-column from Phenomenex (Tecnocroma, Portugal), maintained at 35 °C, was used for chromatographic separation. A gradient elution was performed using a mobile phase (300 µL/min) constituted by a phase A (water/methanol/acetic acid, 94:5:1 (*v*/*v*) and 5 mM ammonium acetate) and a phase B (methanol/water/acetic acid, 98:2 (*v*/*v*) and 5 mM ammonium acetate). The solvent gradient was as follows: 0–7.0 min, 95% A; 7.0–11.0 min, 35% A; 11.0–13.0 min, 25% A; 13.0–15.0 min, 0% A; 15.0–24.0 min 95% A; and 24.0–27.0 min, 95% A. MS/MS acquisition was operated in positive-ion mode with multiple reaction monitoring (MRM), the collision gas was Argon 99.995% (Gasin, Leça da Palmeira, Portugal) with a pressure of 2.9 × 10^−3^ mbar in the collision cell. Capillary voltages of 3.0 KV were used in the positive ionization mode. Nitrogen was used as desolvation gas and cone gas being the flows of 350 and 60 L/h, respectively. The desolvation temperature was set to 350 °C and the source temperature to 150 °C. Dwell times of 0.1 s/scan were selected. The data were collected using the software program MassLynx 4.1. For each analyte, two transitions were selected for identification, and the corresponding cone voltage and collision energy were optimized for maximum intensity. The optimized MS/MS parameters for the target analytes are listed in Table 4.

### 4.9. Method Validation and Quality Control

Linearity was assessed through matrix-matched calibration curves consisting of six calibration points spanning the range from 25 to 400 μg/L (bioaccessibility assays) and from 12.5 to 200 μg/L (bioavailability assays). Precision was evaluated by examining repeatability (intraday precision) and reproducibility (interday precision) of a spiked sample at three different concentration levels. This assessment was conducted using five replicates (with two injections for each replicate) on each precision day.

The limit of detection (LOD) and limit of quantification (LOQ) were established by repeatedly analyzing chromatographic extracts of sample solutions spiked with decreasing amounts of the analytes until signal-to-noise ratios of 3:1 and 10:1 were achieved, respectively.

### 4.10. Statistical Analysis

XLSTAT for Windows 11 Pro version 23H2 (Addinsoft, Paris, France) was used for statistical analysis. The normal distribution of variables was assessed through the Shapiro–Wilk test. Mean comparisons were conducted using two-way ANOVA, with a significance level of 5%. Data are presented as mean ± SD from three independent experiments. GraphPad Prism version 9.3.1 for Windows (Graphpad Software, La Jolla, CA, USA) was employed to generate all graphs.

## Figures and Tables

**Figure 1 toxins-16-00205-f001:**
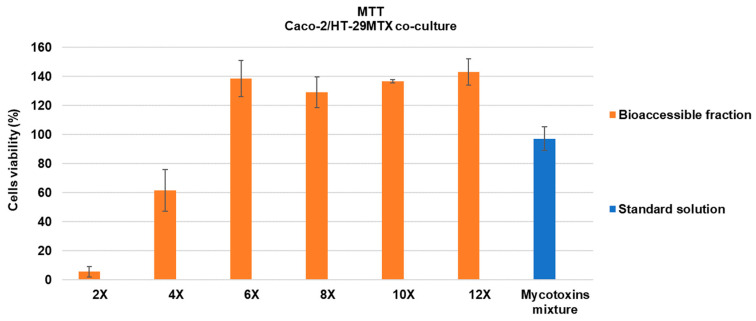
Percentage of cell viability of proliferating Caco-2/HT-29MTX cells, after 180 min of exposure to mycotoxins. Data expressed as mean ± SD of 3 independent experiments (n = 3) against an HBSS control (100% viability). AFB1—aflatoxin B1; ENNB—enniatin B; MIX—mixture of the three mycotoxins; STG—sterigmatocystin. Bioaccessible fraction suffered a heat treatment of 98 °C for 5 min, and was diluted 2, 4, 6, 8, 10, and 12 times. Standard solution: mixture of AFB1 + ENNB + STG of 200 µg/L diluted in HBSS.

**Figure 2 toxins-16-00205-f002:**
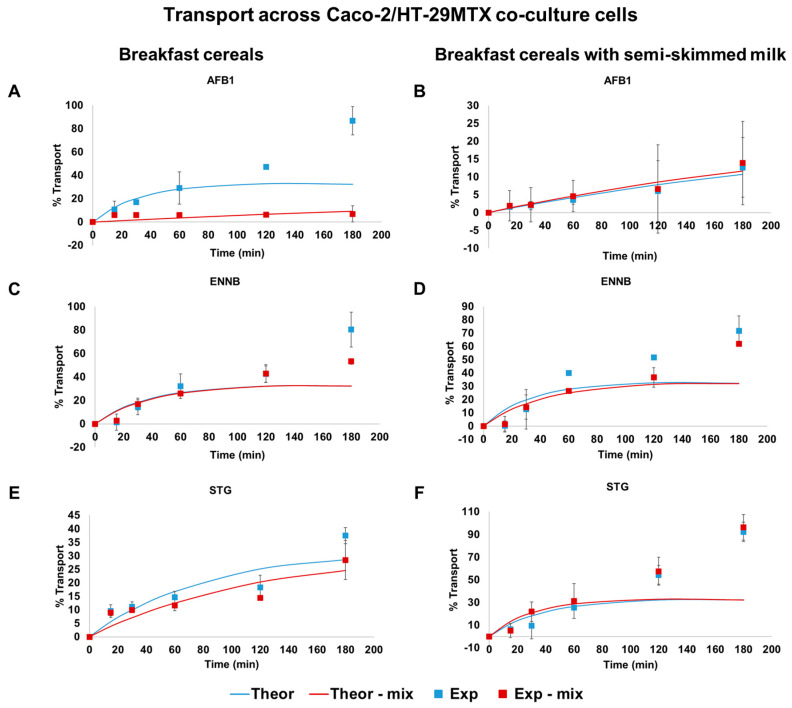
The percentage of AFB1 ((**A**): in breakfast cereals and (**B**): in breakfast cereals with semi-skimmed milk), ENNB ((**C**): in breakfast cereals and (**D**): in breakfast cereals with semi-skimmed milk), and STG ((**E**): in breakfast cereals and (**F**): in breakfast cereals with semi-skimmed milk) transferred to the receiver compartment over 180 min across monolayers of Caco-2/HT-29MTX in the apical → basolateral direction when transported isolated (blue squares and lines) or in mixture (red squares and lines). Data expressed as mean ± SD of 3 independent experiments (n = 3). AFB1—aflatoxin B1; ENNB—Enniatin B; MIX—mixture of the three mycotoxins; STG—sterigmatocystin.

**Figure 3 toxins-16-00205-f003:**
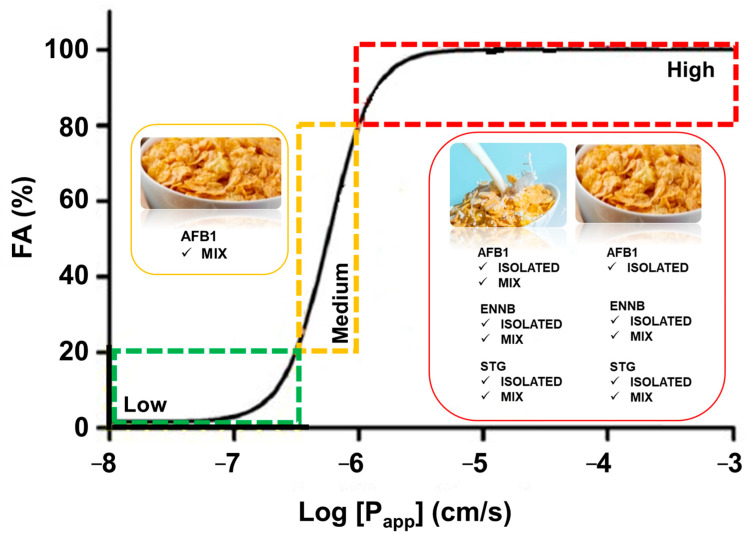
AFB1, ENNB, and STG—isolated and in mixture, in breakfast cereals and breakfast cereals with semi-skimmed milk—positions in the sigmoidal curve according to their FA%—high (red), medium (orange), and low (green) absorption. The sigmoidal curve was built according to (Skolnik et al., 2010 [42]). Data expressed as mean ± SD of 3 independent experiments (n = 3). AFB1—aflatoxin B1; ENNB—enniatin B; and STG—sterigmatocystin.

**Figure 4 toxins-16-00205-f004:**
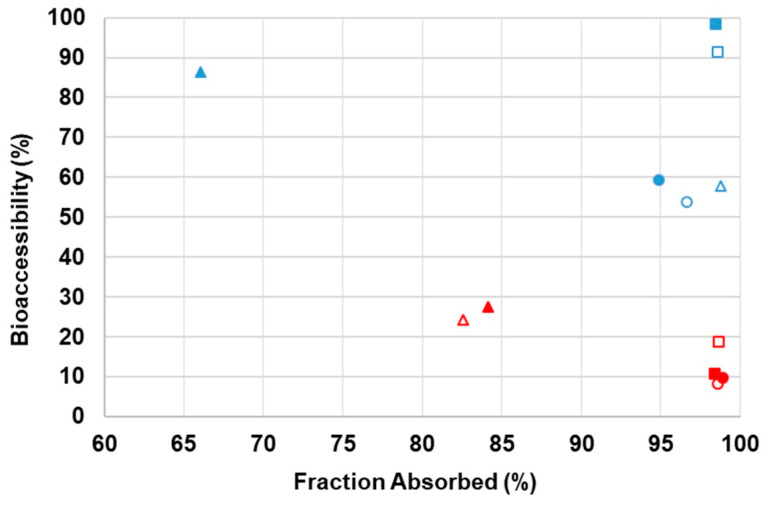
Graphical representation of samples’ bioaccessibility and fraction absorbed. AFB1: triangles; ENNB: squares; STG: circles. Solid triangles, squares, and circles are related to mycotoxin mixture, and empty triangles, squares, and circles are related to isolated mycotoxins. **Blue** color indicates values from samples with breakfast cereals only and **red** with both cereals and milk. Data expressed as mean (n = 3).

**Table 1 toxins-16-00205-t001:** Percentage (%) of bioaccessibility of mycotoxins in cereal and cereal with different types of milk and beverages after gastric and intestinal *in vitro* digestion.

Mycotoxin	*In Vitro* Digestion		Bioaccessibility (%)				
			**BC**	**BCSSM**	**BCSSMLF**	**BCSB**	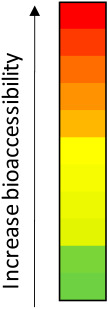
**AFB1**	Gastric	Isolated	<LOD	<LOD	<LOD	<LOD	
		Mixture	<LOD	<LOD	<LOD	<LOD	
	Intestinal	Isolated	57.6 ± 0.1 * ↗	24.2 ± 0.3 * ↗	20.3 ± 1.0 * ↗	3.13 ± 0.17 * ↗	
		Mixture	86.2 ± 1.6 * ↗	27.5 ± 10.3 * ↗	18.1 ± 6.8 * ↗	9.06 ± 4.32 * ↗	
**ENNB**	Gastric	Isolated	49.2 ± 1.6 ^a, b^	47.2 ± 21.6 ^b^	18.1 ± 4.2 ^b^	26.3 ± 6.3 ^a, b^	
		Mixture	52.4 ± 4.5 ^a, b^	40.5 ± 0.3 ^b^	21.6 ± 4.0 ^b^	26.7 ± 8.4 ^a, b^	
	Intestinal	Isolated	91.2 ± 6.4 ^a, b^ ↗	18.5 ± 9.5 ^b^	4.54 ± 1.42 ^b^	21.9 ± 2.2 ^b^	
		Mixture	98.2 ± 7.3 ^a, b^ ↗	10.7 ± 4.2 ^b^	0.62 ± 0.17 ^b^	10.8 ± 0.5 ^b^	
**STG**	Gastric	Isolated	<LOD	<LOD	<LOD	<LOD	
		Mixture	<LOD	<LOD	<LOD	<LOD	
	Intestinal	Isolated	53.6 ± 1.7 ^a, b^ ↗	8.17 ± 0.69 ^b,^ * ↗	19.7 ± 1.7 ^b^ ↗	1.45 ± 0.06 ^a, b,^ * ↗	
		Mixture	59.3 ± 3.7 ^a, b^ ↗	9.52 ± 2.80 ^b,^ * ↗	8.22 ± 1.49 ^b^ ↗	1.97 ± 0.79 ^a, b,^ * ↗	

Abbreviations: BC—breakfast cereal; BCSSM—breakfast cereal with semi-skimmed milk; BCSSMLF—breakfast cereal with semi-skimmed lactose-free milk; BCSB—breakfast cereal with soy beverage. <LOD—below limit of detection. Data expressed as mean ± standard deviation (*n* = 6) for samples following normal distribution. The arrows (↗) indicate an increase in bioaccessibility between gastric and intestinal phases. * Significant difference (*p* < 0.05) between isolated and mixed transport of mycotoxins; different letters in the same row indicate differences (*p* < 0.05) betweenransports.

**Table 2 toxins-16-00205-t002:** Apparent permeabilities (×10^−6^ cm/s) of AFB1, ENNB, and STG isolated and in mixture in the apical → basolateral direction. MB (%) shows the mass balance recoveries of transport experiments and the calculated human fraction absorbed (FA %) for each mycotoxin.

			P_app_ (×10^−6^ cm/s)	MB (%)	FA (%)
**Breakfast cereal**	**AFB1**	Isolated	94.4 ± 0.1	156 ± 16	98.8 *
		Mix	3.31 ± 1.10	28.3 ± 2.0	66.1 *
	**ENNB**	Isolated	83.2 ± 0.0	145 ± 20	98.6
		Mix	78.6 ± 0.0	99.6 ± 2.0	98.5
	**STG**	Isolated	37.6 ± 0.0	72.6 ± 1.9	96.7
		Mix	25.4 ± 0.0	58.7 ± 3.6	94.9
**Breakfast cereal w/semi-skimmed milk**	**AFB1**	Isolated	7.35 ± 0.00	29.2 ± 1.4	82.6
		Mix	8.18 ± 0.00	36.7 ± 2.6	84.2
	**ENNB**	Isolated	91.6 ± 0.0	131 ± 12	98.7
		Mix	72.5 ± 0.0	115 ± 0.3	98.4
	**STG**	Isolated	81.2 ± 0.2	172 ± 11	98.6
		Mix	100.0 ± 0.2	167 ± 18	98.9

Abbreviations: AFB1—aflatoxin B1; ENNB—enniatin B; STG—sterigmatocystin. Data expressed as mean ± SD (n = 3). * significant difference (*p* < 0.05) between isolated and mixed transport of mycotoxins.

**Table 3 toxins-16-00205-t003:** Nutritional properties of the breakfast cereal, milks, and soy beverage used for *in vitro* digestion (as reported in nutritional labels).

Breakfast Cereal Sample	Fat	Fiber	Protein	Carbohydrates
	**(g/100 g)**	**(g/100 g)**	**(g/100 g)**	**(g/100 g)**
Breakfast cereal with corn balls with organic honey	0.6	3.0	5.5	82.0
**Beverages samples**	**(g/100 mL)**	**(g/100 mL)**	**(g/100 mL)**	**(g/100 mL)**
Semi-skimmed milk	1.6	-	3.4	4.9
Semi-skimmed lactose-free milk	1.6	-	3.3	4.9
Soy beverage	1.8	0.5	3.0	2.5

**Table 4 toxins-16-00205-t004:** MS/MS parameters for each mycotoxin under study.

Mycotoxins	Retention Time (min)	MRM Transition (*m*/*z*)		CV (V)		CE (eV)	
		**QIT (*m*/*z*)**	**CIT (*m*/*z*)**	**QIT**	**CIT**	**QIT**	**CIT**
AFB1	15.61	313 > 241.4	313 > 285.3	47	47	36	23
OTA-d5 (IS1)	19.14	409 > 239	409 > 363	32	32	22	22
STG	20.19	325 > 254	325 > 310	35	40	35	25
^13^C_18_-STG (IS2)	20.19	342.7 > 297.4	342.7 > 326.7	24	24	28	28
ENNB	21.63	663 > 218	663 > 336	60	60	70	70

AFB1—aflatoxin B1; CE—collision energy; CIT—confirmation ion transition; CV—cone voltage; ENNB—enniatin B; OTA-d5—ochratoxin d5; QIT—quantification ion transition; STG—sterigmatocystin; ^13^C_18-_STG—^13^C_18_ sterigmatocystin.

## Data Availability

No new data were created or analysed in this study. Data sharing is not applicable to this article.

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
