# Peer review of "In Vitro Digestion and Intestinal Absorption of Mycotoxins Due to Exposure from Breakfast Cereals: Implications for Children’s Health"

_toxins, 2024, doi:10.3390/toxins16050205_

Round 1
Reviewer 1 Report
Comments and Suggestions for Authors
This interesting paper is devoted to the very important and scientifically sound issue - studying intestinal absorption of several mycotoxins, potentially containing in breakfast cereals, and their toxicity. The paper is well written, methods are described comprehensively and can be repeated; English is easy to understand. I believe the manuscript can be accepted for the publication in Toxins. At the same time, I would like to make several comments that would be useful to improve the current version of the MS.
1. The authors should verify the MS text and the Reference list in terms of design. For instance, taxonomic names as well as 'in vitro/in vivo' should be in italic. Also, there could be more information on the key fungal species producing the toxins of interest.
2. From my point of view, it would be appropriate to divide Results and Discussion into two separate sections.
3. I believe more references should be included in the Discussion. For instance, several studies devoted to application of Caco-2 for testing absorption and toxicity of such important mycotoxins as trichothecenes and zearalenone could be added (e.g. Pfeiffer et al., 2011; Bony et al., 2006; Wang et al., 2022; Avantaggiato et al., 2004).
Nevertheless, I am sure the MS corresponds well to the Toxins' requirements and can be accepted for the publication.
Author Response
This interesting paper is devoted to the very important and scientifically sound issue - studying intestinal absorption of several mycotoxins, potentially containing in breakfast cereals, and their toxicity. The paper is well written, methods are described comprehensively and can be repeated; English is easy to understand. I believe the manuscript can be accepted for the publication in Toxins. At the same time, I would like to make several comments that would be useful to improve the current version of the MS. Nevertheless, I am sure the MS corresponds well to the Toxins' requirements and can be accepted for the publication.
- The authors should verify the MS text and the Reference list in terms of design. For instance, taxonomic names as well as 'in vitro/in vivo' should be in italic. Also, there could be more information on the key fungal species producing the toxins of interest.
As requested species names and other Latin words were updated to italic. It is already updated in the manuscript. Addition information about fungal species was added in lines 44-46, 62, 63, 65-68.
- From my point of view, it would be appropriate to divide Results and Discussion into two separate sections.
We appreciate and understand the reviewer's suggestion; however authors would like to keep the present format of the paper (Toxins journal does not require a specific paper organization) since it is our opinion that section 2.1, 2.2, 2.3 and 2.4 should be discussed separately because of the diversity of techniques used and outputs (method performance, bioaccessibility, monolayers integrity and intestinal absorption). The only two sections in which a common discussion would bring new insights into results are sections 2.2 and 2.3. This was already in the manuscript in lines 366-392.
- I believe more references should be included in the Discussion. For instance, several studies devoted to application of Caco-2 for testing absorption and toxicity of such important mycotoxins as trichothecenes and zearalenone could be added (e.g. Pfeiffer et al., 2011; Bony et al., 2006; Wang et al., 2022; Avantaggiato et al., 2004).
References were added to the manuscript in lines 389-391.

Reviewer 2 Report
Comments and Suggestions for Authors
· The manuscript entitled “In Vitro Digestion and Intestinal Absorption of Mycotoxins Exposure from Breakfast Cereals: Implications for Children's Health” has been revised.
· The paper is well-written and includes current references, and the concept is sound.
· The manuscript searched to fill in the knowledge gaps by thoroughly assessing the intestinal absorption and bioaccessibility of three mycotoxins, aflatoxin B1, enniatin B, and sterigmatocystin both singly and in combination when ingested with milk and breakfast cereals.
· The introduction has elements that integrate the work, allowing us to evaluate the context in which the manuscript is inserted.
· The methodology is very clear and suitable for what it proposes to evaluate.
· The results are good and applicable.
· The discussion is good but needs more work.
· The conclusions of the study are based on evidence from the work.
· References are related to the study.
· In general, the presented manuscript represents a quality with good potential for applicability. However, minor comments are suggested below:
Comments:
1. In the Introduction, line 82, please delete “of”.
2. In Figure 2, Line 275, (blue and orange lines) or in mixture (gray and yellow lines). This is not obvious. Please clarify. You can use more clear colors.
3. In Figure 4 legend, Line 385-386, What is the difference between the solid and empty triangles? The same for the squares and circles. Please clarify.
4. Line 440, delete “when not in use.”
5. Line 445, delete “of”.
6. Discussion should be improved to be clearer to the reader
7. Please add reference 2024 if possible.
Comments on the Quality of English LanguageMinor Editing for the English language required
Author Response
The manuscript entitled “In Vitro Digestion and Intestinal Absorption of Mycotoxins Exposure from Breakfast Cereals: Implications for Children's Health” has been revised.
The paper is well-written and includes current references, and the concept is sound.
The manuscript searched to fill in the knowledge gaps by thoroughly assessing the intestinal absorption and bioaccessibility of three mycotoxins, aflatoxin B1, enniatin B, and sterigmatocystin both singly and in combination when ingested with milk and breakfast cereals.
The introduction has elements that integrate the work, allowing us to evaluate the context in which the manuscript is inserted.
The methodology is very clear and suitable for what it proposes to evaluate.
The results are good and applicable.
The discussion is good but needs more work.
The conclusions of the study are based on evidence from the work.
References are related to the study.
In general, the presented manuscript represents a quality with good potential for applicability. However, minor comments are suggested below:
We thank the reviewer suggestions and kind comments.
- In the Introduction, line 82, please delete “of”.
It was done in the manuscript.
- In Figure 2, Line 275, (blue and orange lines) or in mixture (gray and yellow lines). This is not obvious. Please clarify. You can use more clear colors.
Thanks for the correction. The submitted version did, in fact, contain a mistake in the legend. Following the reviewer's recommendation, the color scheme of the figure was simplified for better readability. The figure has been updated in the revised version.
- In Figure 4 legend, Line 385-386, What is the difference between the solid and empty triangles? The same for the squares and circles. Please clarify.
We acknowledge the correction. Solid triangles, squares and circles are related to mycotoxins mixture; and empty triangles, squares and circles are related to isolated mycotoxins. This description was updated in the figure legend.
- Line 440, delete “when not in use.”
It was updated in the manuscript.
- Line 445, delete “of”.
It was updated in the manuscript.
- Discussion should be improved to be clearer to the reader.
We appreciate the reviewer's suggestion and had revised some paragraphs of the results and discussion section increasing paper readability. Sentences revised were changed in the new manuscript version in lines 122-141, 157-184, 185-195, 210-215, 236-273, 276-285, 294, 323-328. We believe that this will make the reading clearer for the reader.
- Please add reference 2024 if possible.
We thank the reviewer suggestion. An updated reference it was added to the manuscript in lines 83, 84.
